# Formative Evaluation of a Home-Based Physical Activity Intervention for Adolescent Girls—The HERizon Project: A Randomised Controlled Trial

**DOI:** 10.3390/children8020076

**Published:** 2021-01-22

**Authors:** Emma S. Cowley, Paula M. Watson, Lawrence Foweather, Sarahjane Belton, Chiara Mansfield, Gabriella Whitcomb-Khan, Isabella Cacciatore, Andrew Thompson, Dick Thijssen, Anton J. M. Wagenmakers

**Affiliations:** 1Research Institute for Sport and Exercise Sciences, Liverpool John Moores University, Room 1.22 Tom Reilly Building, Byrom Street Campus, Liverpool L3 5AF, UK; e.s.cowley@ljmu.ac.uk (E.S.C.); p.m.watson@ljmu.ac.uk (P.M.W.); l.foweather@ljmu.ac.uk (L.F.); c.e.mansfield@2015.ljmu.ac.uk (C.M.); g.whitcomb@2018.ljmu.ac.uk (G.W.-K.); isabellacacciatore@hotmail.co.uk (I.C.); D.Thijssen@ljmu.ac.uk (D.T.); 2School of Health and Human Performance, Dublin City University, D09 Y5NO Dublin, Ireland; sarahjane.belton@dcu.ie; 3Wolfson Centre for Personalised Medicine, Institute of Systems, Molecular and Integrative Biology, University of Liverpool, Liverpool L69 3BX, UK; andrew.thompson@liverpool.ac.uk; 4Radboud Institute for Health Sciences, Department of Physiology, Radboud University Medical Centre, 6525 GA Nijmegen, The Netherlands

**Keywords:** physical activity, adolescents, girls, intervention study, behaviour change, COVID-19

## Abstract

Background. This is a formative evaluation study of the HERizon Project, a home-based multi-component physical activity (PA) intervention for adolescent girls in the UK and Ireland. Although not intended, this study coincided with the initial COVID-19 lockdown restrictions. Methods. A total of 42 female participants, aged 13 to 16 years old (mean = 14.2, SD = 1.1), were randomly allocated to: (i) the HERizon group (*n* = 22) or (ii) the wait-list control group (*n* = 20). Participants in the six-week HERizon group were asked to complete three PA sessions each week and engage in weekly behaviour change support video calls. The primary outcome measure was self-reported habitual PA. Secondary outcomes measures included cardiorespiratory fitness (20 m shuttle run), muscular strength (standing long jump), muscular endurance (push up test), and psychosocial outcomes (Perceived Competence Scale, Body Appreciation Scale, Self-Esteem Questionnaire, Behavioural Regulation in Exercise Questionnaire). Quantitative and qualitative process evaluation data were also collected. Outcome measures were assessed at baseline and after the six-week intervention. Results. There was no significant change in habitual PA between groups (LMM group*time interaction: *p* = 0.767). The HERizon group had significantly increased cardiorespiratory fitness (*p* = 0.001), muscular endurance (*p* = 0.022), intrinsic motivation (*p* = 0.037), and body appreciation (*p* < 0.003) in comparison to the wait-list control group. All participants in the intervention group completed the intervention and compliance to the intervention was high (participants completed 18 ± 2 sessions). Conclusions. Although no change in PA was observed, HERizon resulted in improved physical fitness and psychosocial outcomes. These preliminary findings, alongside positive findings for feasibility and acceptability, highlight potential benefits from the home-based intervention, thus further investigation is warranted.

## 1. Introduction

Physical activity (PA) is associated with multiple health benefits in childhood and adolescence [1,2,3], with the current guidelines recommending individuals under 18 years to engage in an average of 60 min of moderate-to-vigorous physical activity (MVPA) per day across the week [4]. Independent of PA, there is a strong association between cardiorespiratory fitness and adolescent health, including improved metabolic health [5], reduced risk of cardiovascular disease in adulthood [6] and better mental health and academic achievement [7,8]. The government guidelines further recommend children to incorporate activities that develop muscular fitness into their PA, as high levels of muscular fitness in adolescence are associated with improved body composition, self-esteem and skeletal health [9]. Although traditional adolescent physical fitness research has primarily focused on cardiorespiratory fitness, there is a growing interest in assessing the associations between adolescent health and muscular fitness [10,11].

Physical inactivity is a global health concern, with less than 15% of adolescents worldwide meeting the minimum PA guidelines (at least 60 min moderate-to-vigorous physical activity (MVPA) per day) [12]. This issue is particularly prevalent among adolescent girls in the UK and Ireland, with only ~10% being sufficiently active [13,14]. There are numerous interventions aiming to increase adolescent girls MVPA, with the majority being school based [15,16,17]. However, a recent pooled analysis of 17 school interventions found no evidence of improved MVPA, with authors recommending that other intervention contexts should be explored [18]. There appears to be scope to target home and communitys setting for PA interventions as girls typically acquire 50% less MVPA after school hours in comparison to during school [19]. This seems especially pertinent in the current COVID-19 climate as adolescents face self-isolation restrictions and home confinement, alongside school closures and the cancellation of community sport provision [20]. Furthermore, a recent study investigating the effects of COVID-19 on PA levels found that individuals had lower vigorous, moderate and total PA, along with increased time spent sitting in comparison to before the pandemic [21,22]. Few studies have examined interventions to increase participation in MVPA among adolescent girls in home and community settings and further research is needed.

Formative research is a critical step in the design of behaviour change interventions [23]. For example, preliminary qualitative research can optimise randomised controlled trials by improving recruitment rates, help in explaining trial findings and ensures that interventions meet the needs of the target users [24]. Therefore, the first step in the HERizon Project involved conducting formative research using qualitative focus groups with the target population in order to inform the design of a future intervention [23]. The major results of this qualitative study included (1) the importance of having female instructors and mentors, (2) allowing girls to choose from a range of fun physical activities, and (3) giving girls the opportunity to exercise in socially comfortable environment, reducing feelings of self-consciousness [23]. We developed the HERizon Project with the aim of increasing MVPA among adolescent girls in the UK and Ireland. Multiple qualitative studies with adolescent girls have consolidated the above findings in that girls’ perceived feelings of judgement about their bodies, poor self-esteem and low perceived competence are significant barriers to participation in PA [25,26], suggesting that a home-based intervention is a potentially safe and viable option for this population as it reduces the risk of unwanted attention or criticism. Further, recent systematic reviews concluded that interventions that are female only, multi-component and under-pinned by theory are shown to be most effective for increasing adolescent girls participation in PA [27,28]. Therefore, this home-based multi-component PA intervention was created in an attempt to meet the above demands and to add value to this understudied gap in adolescent girls PA literature.

Self-determination theory (SDT; [29,30]) has been drawn upon to understand how to foster motivation towards PA in children and adolescents [31,32]. SDT suggests that motivation exists on a continuum from amotivation (absence of motivation), to controlled regulation (motivation driven by internal or external pressures) to autonomous regulation (self-directed motivation characterised by volition and perceived choice). Controlled motivation involves either external regulation (e.g., engaging in PA for a reward or to avoid a punishment) or introjected regulation (e.g., engaging in PA to preserve the ego, or to avoid feelings of guilt). Autonomous motivation can be either extrinsically regulated (as in identified regulation, working towards a personally important goal; and integrated regulation, where PA is deemed congruent with personal values) or intrinsic in nature (characterised by interest and enjoyment), and is associated with enhanced PA adherence and positive psychological well-being [33]. SDT postulates that for optimal well-being and autonomous motivation, three basic psychological needs must be met—autonomy (the feeling that behaviour is volitional and self-directed), relatedness (a sense of belonging and connectedness to others) and competence (perceived capability of maintaining and enhancing current skills and achieving desired objectives) [34]. Past research has shown that adolescent PA can be improved when these basic needs are met [35,36], thus the HERizon Project intervention was developed utilising the SDT framework.

The primary aim of this study was to conduct a formative evaluation of a home-based PA intervention for adolescent girls MVPA (HERizon Project), informed by previous qualitative work [23] and self-determination theory [29,30]. Formative evaluation is a deliberate assessment of factors that influence an intervention’s progress and effectiveness [37,38] and can provide explanations to the success or failure of certain intervention components [39]. The secondary aim was to assess the preliminary effectiveness of the intervention on physical fitness measures, motivation towards PA, self-esteem, body appreciation and competence. Following the Medical Research Council guidance for developing complex interventions [40], results of the current study will be used to refine the intervention design and inform delivery of a future larger scale trial.

## 2. Materials and Methods

### 2.1. Trial Design

This mixed-methods study was the feasibility phase of a broader intervention of research (The HERizon Project) that aims to develop a theory-based PA intervention targeting adolescent girls. The design was a two-arm randomised controlled trial, comprising (i) the HERizon six-week remote intervention arm and (ii) a wait-list control arm. Block randomisation with country-level (UK and Ireland) stratification was used using Microsoft Excel to allocate the participants on entry. The primary outcome of this study was change in MVPA level. Secondary outcomes included cardiorespiratory fitness, muscular strength and endurance, exercise motivation, perceived competence, self-esteem and body appreciation. Assessments were conducted pre-intervention (April/May, 2020) and repeated immediately post-intervention (June/July, 2020). As this study ran during the COVID-19 pandemic, all participants began the intervention in full national lockdowns, with all local schools and amenities being closed. Restriction began to be lifted in the last week of June in Ireland and in the first week of July in the UK, with some local amenities opening and small outside group gathering being permitted. Due to the nature of this study, participants and project deliverers could not be blinded to the assigned intervention. Ethical approval was granted by the University Ethics Committee [reference: 20/SPS/016] and this study is registered with clinicaltrials.gov [reference: NCT04662775]. Reporting of this study was guided by the Consolidated Standards of Reporting Trials check-list (CONSORT; Appendix A) [41] and the template for intervention description and replication checklist (TIDieR; Appendix A [42]).

### 2.2. Participants and Recruitment

This study aimed to recruit 40 participants, with equal distribution between groups. Adolescents were eligible to take part in this study if (a) they were female, (b) aged 13–16 years, and (c) lived in the UK or Ireland. Girls were excluded if they were pregnant at time of enrolment, were not able to engage in moderate intensity PA or had no access to a smartphone or computer. Parents and participants provided written informed consent/assent prior to randomisation or the collection of any study measurements. Participants were recruited via social media advertisement which invited adolescent girls/parents to express interest.

### 2.3. The HERizon Project

The six-week intervention had two central aims: (1) improve adolescent girls MVPA and physical fitness, and (2) to support participants to adopt and maintain positive physical active habits. The intervention deployed in this study had three components (outlined in Table 1). i. *Exercise sessions:* Participants were asked to complete three 30 min PA sessions each week and record their sessions using a self-report PA logbook. They were given the choice of different types of home-based virtual exercise, e.g., YouTube exercise channels and instructions to design their own workouts. A facilitator for girls participation in PA is to have female exercise instructors and varied, yet challenging, exercises [23]. Therefore, the first author hosted online live group exercise classes each week. This option allowed participants to engage in the social elements of PA but from the comfort and safety of their own homes. It was also thought that having a scheduled set time to join a class may increase motivation and improve adherence. ii. *Behaviour change support calls:* Each participant was allocated an “Activity Mentor” who supported them for the duration of the intervention. The team of 3 Activity Mentors, all trainee sport and exercise psychologists, was supervised by a HCPC-registered Sport and Exercise Psychologist (second author). Participants had seven weekly video calls (an introduction call lasting 30 min and the remaining six calls lasting approximately 10 min; call schedule can be found in Appendix A). Drawing on SDT, video calls employed motivation and behaviour change techniques aimed at fostering autonomy, competence and relatedness in participants [43], e.g., showing unconditional regard, encouraging choice and helping set clear and concrete action plans. Each call was based on a pre-planned session outline and was goal orientated, participant centred and focused on PA (motivation and behaviour change techniques used within the intervention are displayed in Table 1). iii. *No reply SMS:* Participants received three standardised text messages each week, aimed at providing PA-related facts, encouragement and study information, e.g., “did you know physical activity can help improve energy, sleep and mood?” and “Live workouts this week are Monday at 6 pm. Below are sign in details”.

Participants in the wait-list control group were asked to continue their usual PA habits and received no additional contact from the research team outside of data collection points. Following post-intervention data collection, control group participants were invited to participate in the same intervention as described above.

### 2.4. Outcome Measures and Procedures

All outcome measures were collected by participants in their home setting due to COVID-19 restrictions. Prior to baseline measures, participants completed a physical activity readiness questionnaire. The first author had a video call with each participant and a parent/guardian to explain how to conduct all assessments. Parents were asked to supervise all physical fitness tests (cardiorespiratory fitness, muscular endurance, muscular strength) and to input scores into a free, password-protected mobile application (Resistance Training for Teens [44,45]). This application provides visual and written instructions with audio prompts on how to set up, perform and record scores for each test. Parents were asked to verbally encourage participants to push themselves until they could not meet test requirements, e.g., reach the 20 m marker in time for the audio prompt during the 20 m shuttle run, or land on two feet during the standing long jump test. The ground on which participants performed the fitness tests varied between individuals, although within-participant repeat measurements were conducted on the same surface. Participants completed psychosocial questionnaires online via Google Forms on a laptop or mobile device.

#### 2.4.1. Demographic Information

Participant demographic information was collected at baseline which included age, school year, sex, last three digits of their home postcode and ethnicity. Postcodes were mapped against indices of multiple deprivation to estimate socioeconomic status [46,47].

#### 2.4.2. Pubertal Status

A 1-item subscale of The World Health Organisation Health Behaviour in School-aged Children (HBSC) [48] was used to collect menstruation status (“have you begun to menstruate (have periods)?” Yes/No).

#### 2.4.3. Physical Activity and Sedentary Behaviour

MVPA was assessed using the 8-item subscale World Health Organisation HBSC [49] which has been validated with adolescents [49]. This was used to collect self-reported PA and sedentary behaviour over the previous seven-day period at baseline and post-intervention. A mobile phone-based pedometer app (Google Fit) was used to record participants’ steps for seven days at baseline and post-intervention. Steps for the seven days were averaged to calculate a mean daily step score. Girls were instructed to keep their mobile phone on their person as much as possible over the seven-day period, e.g., in their pocket. Mobile pedometers have been validated in adolescents [50].

### 2.5. Physical Fitness

#### 2.5.1. Cardiorespiratory Fitness

Due to national COVID-19 lockdown measures, a home-based measure of cardiorespiratory fitness that was accessible and easy to set up for participants and parents/guardians was required. The 20 metre progressive shuttle run test (20 mSRT) was used to provide an estimate of cardiorespiratory fitness [51]. Participants ran back and forth between two markers positioned 20 m apart measured by a tape measure, at a pace signalled by an audio file on the Resistance Training for Teens (RTT) app. The test starts at a pace of 8.5 kmph and gradually increases in 0.5 kmph increments as the levels progress (pace increases approximately every minute). The test ended when the participant could not make it to a line for two consecutive beeps and the final successful stage was recorded using the app. Parents were instructed to provide verbal encouragement throughout the test with the aim of girls reaching volatile exhaustion. This test has been validated with adolescents previously [52].

#### 2.5.2. Muscular Endurance

A 90 degree push up test was used as a measure of upper body muscular endurance. Using an audio file on the RTT app, participants performed push ups at a cadence of 40 bpm. Participants began in a high plank position (hands positioned under the shoulders, toes touching the floor, back in a straight line) and lowered themselves to the ground in a controlled manner until their arms are at a 90 degree angle from which they then pushed back up and returned to a high plank position. Parents counted and recorded the number of push ups and the test was terminated if participants could not maintain correct exercise form or voluntarily stopped. The push up test has been validated as a measure of muscular endurance with adolescents [53].

#### 2.5.3. Muscular Strength

A standing long jump test was used as a measure of lower body muscular strength. Participants began standing with their toes behind a starting marker and then performed a long jump, jumping from and landing on two feet. A parent/guardian measured the distance travelled from starting marker to the participant’s foot in the landing position using a tape measure. Participants completed this measure twice with the longest jump being recorded as the final score. The long jump test has been found to be a valid and reliable measure of adolescents’ muscular strength [54].

#### 2.5.4. Psychosocial Questionnaires

Exercise motivation was measured using the Behavioural Regulation in Exercise Questionnaire 3 (BREQ-3) which combines 19-items from the BREQ-2 [55], four additional integrated regulation items (validated by Wilson et al. [56]), and one additional introjected regulation item [57]. The questionnaire contains six subscales, ranging from amotivation, through controlled motivation (external regulation, introjected regulation) to autonomous motivation (identified regulation, integrated regulation, intrinsic motivation). Participants were scored using a five-point Likert scale ranging from “Not true (0)” to “True (4)”. Cronbach’s alpha was used to assess the internal consistency of the BREQ-3 subscales. With the exception of amotivation (0.61 to 0.70) and identified regulation (0.76 to 0.84), the internal consistency in our sample was poor (Cronbach’s alpha between 0.45 and 0.59 for external regulation, 0.26 to 0.42 to introjected regulation, 0.14 to 0.47 for integrated regulation and 0.10 to 0.44 for intrinsic motivation).

Body image was measured by the 10-item Body Appreciation Scale [58] which was also scored using a five-point Likert scale ranging from “Never (0)” to “Always (4)”. Questions include statements such as “I respect my body” and “I feel love for my body”. Higher scores reflect a higher level of body appreciation. Responses demonstrated excellent internal consistency (Cronbach’s alpha between 0.94 and 0.95).

PA competence was measured by an adapted 4-item Perceived Competence Scale [59] which was scored using a seven-point Likert scale ranging from “Strongly disagree (1)” to “Strongly agree (7)”. Questions include ”I am capable of being physically active regularly”. Higher scores indicate a higher perceived competence in PA. Responses within our sample demonstrated excellent internal consistency (Cronbach’s alpha between 0.85 and 0.91).

Finally, self-esteem was measured by the 12-item Adolescent Self-Esteem Questionnaire [60], which was scored using a five-point Likert scale ranging from “Almost all of the time (1)” to “Hardly ever (5)”. Questions include “I feel I can be myself around other people”. A higher score indicates a higher level of self-esteem. To generate an overall score for each questionnaire, questions were scored using the corresponding Likert scale and a mean score was then calculated. Responses within our sample demonstrated acceptable internal consistency (Cronbach’s alpha between 0.68 and 0.72).

### 2.6. Process Evaluation

Process evaluation took mixed-methods data from a number of different sources throughout the intervention to assess reach, implementation, adherence and acceptability (see Table 2). Data included screening questionnaires, participant PA logbooks, Activity Mentor call logbooks, semi-structured interviews and pre/post-intervention outcome measures.

To measure implementation of planned intervention activities (dose delivered), the Activity Mentors kept a weekly log of all video calls, including tracking attendance and duration of calls. The exercise instructor for live workouts kept a record of all sessions provided, including date and time.

Adherence to physical activity sessions by participants was monitored using a hard-copy self-report logbook and through interviews. During weekly video calls with Activity Mentors, participants set an action plan of what PA sessions they would complete in the coming week by writing into their logbook what day, time and type of activity they would do. Participants were instructed to tick off each session in their logbook once it had been completed. These logbook had a dual purpose; as an intervention behaviour change strategy, as well as a research measure to collect participant adherence to PA sessions. Participants received a weekly text message reminder to complete their PA logbook. During the following video call with their Activity Mentor, participants would show their mentor their logbook and discuss their PA sessions. Mentors kept a record of how many sessions participants completed each week. PA sessions were summed at the end of the six-week intervention as a measure of participant adherence.

### 2.7. Interviews

Semi-structured phone interviews (*n* = 10) were conducted after post-intervention measures with a purposely selected subsample of participants from the intervention group of ranging adherence levels, identified from self-reported logbooks. Discussions lasted between 20 and 40 min and were focused on participants’ perceptions of the intervention components, including recruitment processes, perceived benefits of the intervention and facilitators and barriers to intervention adherence (interview guide can be found in Appendix A). Interviews intervention conducted by the lead author (a female PhD candidate with MSc qualification and experience conducting qualitative studies), who also instructed the live group workout sessions.

### 2.8. Sample Size

A sample size calculation for effectiveness is not applicable given this is a formative assessment. However, we do acknowledge the need to ensure we have enough participants in order to answer important questions about the feasibility of future investigations (e.g., recruitment and retention rates) and being able to provide statistical point estimates. A sample size of 40 would give acceptably precise estimates of the variability in changes in outcome variables. For example, for outcomes measured as a proportion, e.g., the 95% confidence interval (modified Wald method) for a drop-out rate of 20% would be 10% to 35%.

### 2.9. Data Analysis

Between-group differences at baseline were explored using independent t-tests for continuous variables and chi-square or Fisher’s exact test for categorical variables Two-factor (group*time) linear mixed models were used to assess the effect of the six-week intervention between groups on outcome variables. Age and deprivation status were entered into the models as covariates. The covariance structure for each model was determined using a chi-square distribution, with the selection determined by the most parsimonious structure that optimised model fit. The least significant difference approach to multiple testing was adopted given this was a feasibility study with a small sample size. Normality of residuals from the linear mixed models were assessed using Q-Q plots, and all variables satisfied this assumption except internet use, which was log10 transformed and reanalysed.

Descriptive statistics are presented as the mean ± SD, unless stated otherwise, and outcomes of linear mixed models as the mean (95% CI). Statistical analysis was performed using SPSS for Mac (Version 26.0, SPSS, Chicago, IL, USA) and statistical significance was set at *p* < 0.05.

As outlined above, semi-structured interviews were used as a data source for the process evaluation aspects of the intervention (Table 2). Interviews were audio recorded and transcribed verbatim by the first author. Data were imported into NVivo 12.0 and a reflective thematic analysis was undertaken to identify themes falling within each of the a-priori process evaluation research question [62]. Preliminary analysis was conducted by the first author, who conducted the interviews, and to enhance rigor, three researchers (second, third and fourth authors) acted as ‘critical friends’ by reviewing sections of raw data and meeting regularly as a group to discuss and debate the initial thematic structure and provide alternative views of the data [63].

## 3. Results

### 3.1. Reach

Sixty-two participants were invited to take part in this study and forty-two consented (68% response rate, Figure 1). Participants were recruited via social media advertising and subsequent “snowball” effects (i.e., sharing of information between friends).

All 42 participants (*n* = 42) provided complete data at baseline and post-intervention assessments (Figure 1). Descriptive and baseline characteristics for both groups are displayed in Table 3. There were no significant differences in demographic details between groups. Baseline fitness measures were generally similar between groups, although the intervention group had shorter long jump scores than the control group (162 ± 23.0 cm vs. 171 ± 40.5 cm, *p* = −0.045). PA and sedentary behaviours were also similar except the intervention group spent significantly more hours playing video games (0.95 ± 0.2 h vs. 0.85 ± 0.3 h, *p =* 0.021 and the control group engaged in significantly more vigorous PA per day (2.41 ± 0.2 h vs. 0.85 ± 0.4 h, *p* = 0.044).

### 3.2. Outcome Evaluation

Participant MVPA, physical fitness and psychosocial questionnaire results are displayed in Table 4. No significant difference in MVPA was found between groups following the intervention (*p* = 0.767). There were significantly greater improvements in 20mSRT (1.8 vs. 0.3 stages; *p =* 0.001) and push up score (6.9 vs. 2.3 repetitions; *p =* 0.022) in the intervention group compared to the control group. Intrinsic motivation was significantly greater in the intervention group in comparison to the control group, where intrinsic motivation declined (0.16 vs. −0.16; *p =* 0.037). Body appreciation scores were also significantly improved in the intervention group in comparison to the control group (0.44 vs. −0.11; *p* < 0.003). No significant between group*time effects were observed for long jump, daily mean step count, vigorous physical activity, sedentary behaviour, competence, self-esteem, amotivation or external, introjected, identified or integrated regulation.

Interview data, which collected girls’ perceived impacts of this study, align with these quantitative results (see Table 5 for participant illustrative quotes). Participants felt fitter after the six-week intervention and were proud to see improved 20mSRT scores. Participants also commented on feeling more confident and comfortable in their bodies and abilities, as well as enjoying physical activity more and engaging in it without pressure from parents.

### 3.3. Implementation

Activity Mentors hosted 154 behaviour change support video calls (seven calls per participant). Calls were planned to be 15 min each and actual delivered call durations were 9 ± 3.0 min (mean duration for calls 1 to 6). Total mean duration of call time of 69 ± 22.2 min per participant (mean call duration of 18 ± 5.4 min for the first introduction call. 396 non-reply SMS messages were planned to be sent during the intervention (three messages per participant per week). This was accomplished. Eighteen live group exercise sessions were planned for (three per week) during the intervention and all sessions were delivered via password-protected Zoom link (https://zoom.us).

Participants enjoyed the structure of having calls with their Activity Mentor at the same time each week and thought that the short duration was convenient and acceptable.


*Activity Mentor calls were really good, they were really short and simple and the same every week and they were at a good time as well like 5pm on Mondays [P1].*


Some participants said that they did not receive any SMS as they did not check their text messages; for others, it was due to the research team using incorrect phone numbers. For girls who did receive messages, the content, timing and frequency was acceptable and many cited that they were helpful reminders.


*The texts were good because it wasn’t like annoying or too many, it was more like just reminders and not all the time [P16]*


The girls who tried the live workouts expressed positive feedback and noted that having a set timetable for the sessions helped them to plan and adhere to their PA for the week. The short duration was also positively received by the participants.


*I liked the length of the live workouts like 20 to 30 min because it’s not that long and you can do them anywhere like in your room which is good I liked that [P1].*


### 3.4. Adherence

Participants were asked to complete 18 PA sessions during the six-week intervention. According to self-report logbooks, median and inter-quartile range for adherence was 18 ± 2 sessions. Of the seven behaviour change support calls, mean attendance was 6.0 ± 0.3. All participants in the intervention group completed the six-week intervention. Table 5 outlines illustrative quotes of participants perceived barriers and facilitators towards intervention adherence.

Based on interviews, participants appear to have enjoyed the core components of the intervention and listed many facilitators to their adherence. For example, the live workouts were popular with many participants saying they enjoyed the variety and novelty of the exercises and even enjoyed that some were challenging. It was important to the girls that the sessions were not monotonous or boring and some also spoke of the importance of having a relatable instructor who they could see was also sweating and out of breath. Girls from all interviews commented on the importance of being given the choice in what type of PA they engaged with, and that having a weekly logbook to keep track of their PA sessions was helpful for creating a routine. Having support was a key facilitator to girls’ adherence. Engagement was often helped by parents or siblings doing PA with them, and by knowing other girls in their age group were also taking part in the intervention. Activity Mentors played a vital role in support provision with many participants explaining that they would not have stuck to the intervention if it were not for their weekly calls. Some girls said that they felt the calls were a positive and encouraging environment as they were not told off for missing a PA session. As the intervention ran during the initial lockdown of 2020, many of the girls commented that it was nice to have someone to speak to each week as they sometimes felt lonely/isolated being at home.

Barriers to adhering to three PA sessions each week included period pain and times of the live workouts not suiting participants’ schedules. Many of the girls suggested adding more options to the PA menu and to make it clear for future participants that they are allowed to do activities not listed on the menu, e.g., cycling or gymnastics. Some participants said that they felt uncomfortable exercising as part of an online group where others would see them and therefore they avoided the live sessions. These participants all recommended making it clear to future participants that they can still join the group sessions whilst keeping their cameras turned off as this may increase attendance.

### 3.5. Acceptability of Outcome Measures

Participants reported that the field fitness tests were easy to set up and input on the Resistance Training for Teens mobile app. Questionnaires were completed by girls on their mobile phones and were found to be acceptable as they did not take too much time and did not require them to type, only to check boxes.

## 4. Discussion

This study, which took place during the COVID-19 pandemic lockdown restrictions, aimed to assess the feasibility, acceptability and preliminary effectiveness of a six-week, home-based PA intervention for adolescent girls using a mixed-methods formative evaluation. Compared to a wait-list control group, the intervention group significantly improved in 20 m shuttle run test and push up scores, intrinsic motivation and body appreciation scores. The findings of this formative evaluation add to the limited evidence base concerning adolescent girls PA interventions in the home and community settings.

### 4.1. Preliminary Effectiveness

Although there was a significant improvement in physical fitness scores and intrinsic motivation in the intervention group compared to the control group at post-intervention, there were no significant differences in MVPA compared to the control group. This may be due to multiple factors including the use of self-report questionnaires and the length of the intervention. Subjective measures of PA typically record participants perceived PA levels, rather than actual PA levels [64], with girls being more likely to overestimate their PA using self-report measures than boys [65] Other self-report inaccuracies may include recall issues [66] and knowledge around what constitutes as PA, as many adolescents view physical activity as being synonymous only to sport and structured physical education lessons [67]. Both the intervention and the control group saw significant increases in daily step count (2230 for the intervention group and 2085 for the control group). This further suggests that the self-report measure of PA is not capturing the changes in MVPA. Interventions lasting at least 12 weeks are found to have the greatest effect on adolescent girls MVPA [29,68]. Other studies with medium- to long-term follow up have also seen further increases in MVPA and improvements in psychosocial measures [69]. For example, one study found even greater improvements in MVPA, goal setting and self-efficacy at 9 month follow up in comparison to post-intervention [70]. While there was no change in physical activity, there was a significant change in the determinants of PA, such as intrinsic motivation, which may lead to changes in physical activity over the longer term. Therefore, future trials of HERizon should use objective measures of PA, be at least 12 weeks and have a follow-up data collection point.

High cardiorespiratory fitness in youth is associated with lower rates of obesity and prevalence of metabolic syndrome in adulthood [68]. The intervention group showed significant improvements in cardiorespiratory fitness following the intervention, with a mean increase that was 1.5 stages greater in the shuttle run test compared with the control group. This converts to an estimated mean VO2max improvement of 6.9 (18.6%) for the intervention group and a 0.7 (1.9%) improvement for the control group. A meta-analysis of over 9000 children reported that girls with a VO2max of less than 35 mL.kg.min had 3.6 times greater likelihood of having cardiovascular disease [69]. This suggests that the intervention may have clinical benefits as baseline scores for the intervention group were 30 mL of O_2_ per kg body weight per min but increased to 37.1 following the six-week intervention. During interviews, participants said that they enjoyed cardio-based exercise more following the intervention than when compared to baseline, with many explaining that this increased enjoyment was due to feeling fitter, making the exercise easier. A study conducted with female university students found similar results with participants perceived physical fitness being a significant predictor of exercise enjoyment and adherence [71]. During interviews, many girls said that they intended on maintaining their fitness after the intervention by going on regular runs with a parent or friend, and some had even signed up to local 10 km races.

A recent meta-analysis concluded that muscular fitness during adolescence was negatively associated with obesity and cardiometabolic conditions in adulthood and positively associated with bone health [10]. However, the literature lacks studies that include strength-based measures in adolescent interventions. Girls in the intervention group improved muscular endurance by 7 push up repetitions (31.8%) at post-intervention, with no change for control participants. Although the intervention was relatively short, some girls said that they saw positive improvements in their body composition, in particular increases in muscle mass in the lower body and abdominals. Many girls also said that they felt stronger and were proud to see improvements in strength. Past qualitative work has found many girls do not want to develop muscle mass as they are fearful of being perceived as masculine and therefore they often disengage with PA [72,73,74]. Contrary to such literature, improvements in strength and muscle gain were seen as a positive outcome by participants in the current study. These results are reflective of the Project WONDER trial [75], which found that both active and inactive adolescent girls perceived resistance training to be enjoyable, beneficial and something that helps to relieve stress. In the current study, there was a non-significant increase in the result of the long jump test (improvement of 7 cm for the intervention group, decrease of 1 cm for the control group, *p* = 0.193) that we used as a measure of muscle strength in this home-based study. As the intervention included a wide range of physical activities varying in intensities, it is also difficult to conclude what specifically improved the fitness measures. The intervention group baseline 20 mSRT scores were reflective of the lower 5th percentile of the NHANES 1999–2002 study with over 3300 12–18 year olds [76], suggesting that any form of PA may be beneficial in improving cardiorespiratory fitness in youth when baseline fitness is so low. Furthermore, statistical artefact created by regression to the mean needs to be considered in such situations. Push up scores are reflective of global average [77]. However, the significant improvement may be due to the majority of PA options having some element of resistance training, and therefore it is likely that individuals experienced more exposure to muscle strengthening activities during the intervention than at baseline.

Low self-worth and body dissatisfaction are common barriers to girls participation in PA [78,79] and given the high prevalence of negative body image among adolescents in the UK and Ireland [80,81], the positive changes found in participants body appreciation scores following the HERizon intervention are promising. Supportive and caring environments that foster a culture of acceptance of individual differences can reduce body dissatisfaction [82], and the HERizon intervention aimed to create such an environment through giving girls the autonomy to choose the PA they participate in, providing them opportunities to engage with other girls on the intervention, and through individual needs-supportive mentoring. These components may have also helped to improve participants intrinsic motivation scores, which were significantly greater in the intervention group compared to the control at post-intervention. During interviews, girls commented on enjoying the activities, not feeling pressured to be active and that they felt supported during weekly calls with their Activity Mentor. A recent SDT-based intervention for adolescent girls with overweight and obesity found that similar psychosocial support led to improvements in physical self-worth, body satisfaction and perceived physical conditioning [83]. As girls with high body appreciation are more likely to engage with PA [84], it may be important to implement similar psychosocial supportive components in future trials with girls who are inactive.

### 4.2. Feasibility/Acceptability

This intervention targeted adolescent girls PA habits by implementing a number of needs-supportive motivational and behaviour change techniques, as MVPA has been shown previously to be positively influenced by autonomous motivation [85]. There was an extremely high retention rate to this study, with all participants, in both intervention and control groups, completing all baseline and post-intervention outcome measures. Participants in the intervention group also showed high adherence to the PA programme with self-report logs showing that girls completed 18 session over the six-week period (the target was 18 sessions, 3 sessions per week). Each participant was assigned an Activity Mentor who worked with them throughout the intervention, supporting their adherence to the intervention via weekly video calls and by helping participants develop goal setting, action planning and coping skills. Previous mentoring interventions with adolescents have shown positive results, including improvements in MVPA [35], self-confidence and self-efficacy [86,87]. Participants reported that the duration, content and frequency of the calls were acceptable. Further, many girls noted that these regular catch-up calls were important for not only keeping them accountable, but also to help them feel connected to others by having someone to talk to each week. This seems particularly relevant given the intervention ran during the strict initial COVID-19 lockdown in the UK and Ireland when schools were closed. It is well documented that the support of peers, coaches, teachers and family can play a vital role in facilitating girls PA participation [88,89] and the Activity Mentors appear to have helped fulfil participants need for relatedness within the intervention.

Girls in the intervention group were asked to complete three PA sessions each week and were provided with a “Physical Activity Menu” of sample ways to be active at home, without requiring equipment or much space, which included links to YouTube exercise videos and instructions on how to create their own workout routines. From qualitative interviews, the participants responded positively to being given the independence to make their own PA choices. Typically, research studies require all participants to adhere to a uniform PA intervention [90,91,92]. However, results of the current study suggest that a more flexible approach has the potential to improve participants autonomy and intrinsic motivation.

The live workout sessions were received particularly well by participants who engaged with them (50% of participants in the intervention group took part in at least 1 of the live workout sessions). The sessions lasted approximately thirty minutes and consisted of various types of interval training, including strength-based exercises, Pilates and high-intensity interval training. Modifications were given for each exercise and girls were encouraged to assess their current skill, fitness and energy level and adapt exercises when necessary, further giving them an opportunity to be autonomous. During interviews, girls commented that they liked being challenged by the sessions, as well as the variety/novelty of exercises. Past studies that have used similar individual, non-team-based activities, with a focus on personal improvement rather than competition, resulted in significant improvements in MVPA [93,94]. Another potential reason for the popularity of the live sessions may be due to them providing an opportunity for girls to interact with others of their own age. Although girls were not physically together, and most kept their computer cameras turned off during the exercise, there were opportunities throughout the sessions for group interaction, e.g., during the warm up and via the chat box function. Past work investigating the effect of virtual group exercise in this population is limited. However, a Facebook mHealth intervention with adolescent cancer survivors found positive changes in social functioning following a 10 week remote intervention [95]. Another virtual group exercise intervention with adults found significant improvements in life satisfaction, with social interaction being positively associated with group cohesion [96] suggesting that individuals do not necessarily need to be physically together to get social benefits from group exercise. The intensity of the workouts was a common topic of conversation during these live sessions, with some girls commenting during the post-intervention interviews that sweating alongside others their age and being able to relate to their struggles was a facilitator for their commitment to the intervention. When designing an ad campaign to promote PA to adolescents, it was found that it was not celebrities who were viewed as being the most influential to teenagers [97]. Instead it was relatable people who share their struggles, e.g., body image concerns. Other recent qualitative research found that girls were not motivated to exercise by “air-brushed” role models and instead wanted to see the instructor out of breath and red faced too [23]. Although being untidy and sweaty is a commonly cited barrier for girls PA ([25], the protection of exercising through a computer screen seems to overcome this.

### 4.3. Strengths and Limitations

Due to the limited research conducted outside of the school setting, this multi-component intervention, with a robust and feasible study design, is novel in its remote home-based approach to increase adolescent girls PA. The intervention appears to be acceptable and feasible to this cohort as adherence to PA sessions and Activity Mentor calls were high (86% and 94% adherence respectively), with no participants dropping out of the intervention during the six weeks. This also highlights the potential value of collecting outcome measures remotely as compliance was high, with a complete data set at post-intervention, and participants had positive experiences of conducting measures at home, thus minimising participant burden of attending laboratories or difficulties collecting measures within a school setting. Furthermore, we have limited knowledge regarding PA engagement during epidemics, and this study provides initial insight for a group considered at high risk of both low PA and mental health problems. Although the forced national lockdowns caused by COVID-19 are extreme, appropriate consideration needs to be given for maintaining health if such policies are enacted again for subsequent global events.

Despite the study strengths, several limitations must be acknowledged. The intervention had a relatively small sample size. However, it is acceptable given that this study’s primary aim was to assess feasibility and explore preliminary effectiveness. While results are promising, a future definite trial is needed for confirmation of improvements in outcomes. Physical activity assessment was limited due to the subjective measure of self-report questionnaires which can often result in overreported physical activity and under-reported sedentary behaviour [98]. It is also acknowledged that participant’s daily step count may have been inaccurate if they did not carry their phone on their person for the duration of the day. For a more objective evaluation of HERizon’s potential impact on adolescent MVPA, accelerometers should be used in a future trial. Further, fitness measures were conducted remotely without the supervision of trained researchers and may have been completed in non-standardised conditions [99]. Other fitness test limitations include between-participant variation in the surfaces used for the 20 m shuttle run test and standing long jump [100]. However, individuals completed baseline and post-intervention measures on the same surface, thus the impact on within-subject variability and model estimates is likely to have been limited. Future trials should consider the use of the Chester Step Test [101] as a home-based measure of cardiorespiratory fitness, as due to the lockdown restrictions at the time of the trial many participants found it difficult to find a suitable area to perform the 20 m shuttle run test. Although the 20 m shuttle run test is considered a gold-standard measure of cardiorespiratory fitness [102,103], the Chester Step Test has been validated among adolescents [104]. Finally, the primary role of the first author was project management (including recruitment, collecting outcome measures and conducting qualitative interviews). However, she also conducted all live workout sessions. In recognising the potential influence this might have on interview data, we made efforts to ensure participants felt at ease and felt able to share any potential negative views. We also involved “critical friends” (who were not intervention deliverers) in the interview analysis to ensure different perspectives were considered.

## 5. Conclusions and Future Direction

Results of this study provide evidence for the potential benefits of a home-based PA intervention on improving adolescent girls physical fitness, intrinsic motivation and body appreciation. Home-based outcome assessments were feasible and acceptable for participants, and although these improvements are promising, it must be acknowledged that this study ran during an unprecedented time of the initial COVID-19 lockdown restrictions and therefore further investigation during more typical conditions is needed.

Integrating qualitative and quantitative data within process evaluation was valuable as it provided insight and further explanation to the successes and failures of intervention components. This has prompted reflection and refinement for future recruitment, delivery and assessment strategies. Due to the intervention implementing multiple behaviour change strategies, e.g., live group workouts, participant logbooks, text messaging and mentorship, it will be important that future trials investigate the active intervention components and their corresponding effect on MVPA, physical fitness and psychosocial scores. Effectiveness can be more thoroughly assessed through a longer-duration intervention, with a larger sample size and a medium- to long-term follow up.

## Figures and Tables

**Figure 1 children-08-00076-f001:**
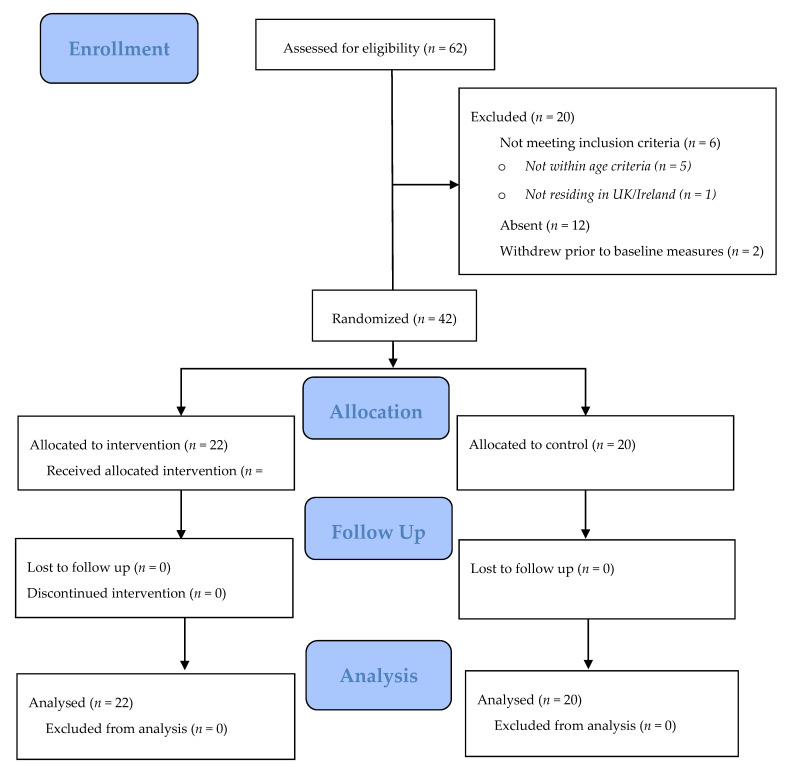
Flow chart of participants through this study based on the CONSORT 2010 flow diagram.

**Table 1 children-08-00076-t001:** Intervention components and behaviour change techniques.

Intervention Component	Dose	Description	Basic Psychological Need(s) ^a^	Motivation and Behaviour Change Techniques ^b^
PA menu	6 weeks	Participants were asked to complete 3 × 30 min PA each week. Participants received a list of suggested PA options they could choose from, including weblinks to home-based YouTube videos and instructions on how to create their own workout plans.	Autonomy	MBCT5: Provide meaningful rationale MBCT6: Provide choice
Text messaging	3 × week (6 weeks)	Participants received three text messages per week that provided information on the live workout schedule, reminders to complete their PA logbooks, and general PA encouragement.	Relatedness	MBCT13: Providing opportunities for ongoing support
Live workout sessions	3 × 30 min × week (6 weeks)	Researcher-led live group workouts were offered but were not compulsory. Workouts were body weight circuit training requiring little space or equipment delivered via Zoom.	Autonomy	MBCT3: Use non-controlling information language MBCT6: Provide choice
			Relatedness	MBCT10: Show unconditional regard MBCT11: Show interest in the person
			Competence	MBCT17: Assist in setting optimal challenge
Individual support calls	1 × 30 min 6 × 10 min	Participants received weekly video calls from their allocated Activity Mentor. These calls were aimed to be a source of behaviour change support and accountability. Calls were based on a pre-planned topic structure (found in Appendix A).	Autonomy	MBCT1: Elicit perspectives on condition or behaviour MBCT2: Identification of sources of pressure for behaviour change MBCT3: Use non-controlling information language MBCT5: Provide meaningful rationale
			Relatedness	MBCT10: Show unconditional regard MBCT11: Show interest in the person MBCT12: Use empathic listening MBCT14: Prompt identification and seek available social support
			Competence	MBCT15: Address obstacles for change MBCT18: Offer constructive, clear and relevant feedback MBCT19: Help develop a clear and concrete plan of action
PA logbook	6 weeks	Participants were asked to keep a hard-copy log of their PA sessions each week and record what types of activity they did in each session. This was reviewed weekly by their Activity Mentor. Self-reported PA sessions recorded in the logbook were taken as a measure of participant adherence to the PA intervention.	Autonomy	MBCT6: Provide choice
			Competence	MBCT0: Promote self-monitoring

^a^ Although some strategies support multiple psychological needs, only the primary strategy/s are listed. ^b^ MBCT: motivation and behavior change techniques [43].

**Table 2 children-08-00076-t002:** Data sources used to assess the implementation of The HERizon Project.

Data Source	Sample	Date of Data Collection	Implementation Aspect Assessed				
			*Reach*	*Implementation*	*Adherence*	*Impact*	*Acceptability*
Screening questionnaire	42 adolescents	Baseline	X				
Activity Mentor call logbook	3 mentors	Weekly		X	X		
Participants PA Logbook	42 adolescents	Weekly			X		
Live exercise session logbook	1 instructor	Weekly		X			
Interviews	10 adolescents	Post-intervention	X		X	X	X
Outcome measures	42 adolescents	Baseline and Post-intervention				X	X

Implementaion of the intervention was assessed using a modified version of the RE-AIM framework [61]. Abbreviations: PA physical activity.

**Table 3 children-08-00076-t003:** Baseline demographics of study participants by group.

Characteristics	Intervention (*n* = 22)	Control (*n* = 20)	*p*-Value
Age, mean (SD), years	14.0 (1.2)	14.3 ± 0.9	0.484
School year (SD)	9 (1.4)	9 ± 1.0	0.330
Ethnicity, *n* (%)			
White	22 (100%)	19 (95%)	0.149
Reside in UK	17 (77%)	11 (55%)	0.126
Socioeconomic status, *n (%)* ^a^			0.212
Tercile 1 (most deprived)	9 (41%)	11 (55%)	
Tercile 2	8 (36%)	8 (40%)	
Tercile 3 (least deprived)	5 (23%)	1 (5%)	
Menstruation status ^b^			
Yes, %	18 (82%)	15 (79%)	0.817

*t*-tests were used to establish differences between groups for outcomes variables at baseline (significance set at *p* < 0.05). *Abbreviations:* UK United Kingdom. ^a^ Socioeconomic status determined by population tercile using UK Index of Multiple Deprivations and The Pobal HP Deprivation Index based on home postcode (1 = most deprived, 2 = median deprived, 3 = least deprived). ^b^ One participant did not report menstrual status.

**Table 4 children-08-00076-t004:** Outcomes for physical fitness, physical activity and psychosocial measures in the intervention and control groups.

Variable	Baseline (Mean ± SD)		Post-Intervention (Mean ± SD)		Adjusted Mean Difference between Time Points (95%CI)		Group * Time *p*
	Intervention	Control	Intervention	Control	Intervention	Control	
**20m shuttle run, stages**	5.22 ± 2.6	7.04 ± 4.2	7.04 ± 3.3	7.34 ± 4.3	1.82 (1.20, 2.43) ***	0.31 (−0.33, 0.96)	0.001^†^
**Long jump, cm**	162 ± 23.0	171 ± 40.5	169 ± 30.2	170 ± 42.1	6.7 (−1.4, 14.8)	−1.0 (−9.5, 7.5)	0.193
**Push ups, repetitions,**	15 ± 11.3	15 ± 15.41	22 ± 12.5	15 ± 12.73	6.9 (4.3, 9.5) ***	2.3 (−0.5, 5.2)	0.022 ^†^
**MVPA, minutes**	3.62 ± 2.4	4.35 ± 1.76	3.91 ± 1.8	4.37 ± 1.92	0.25 (−0.70, 1.2)	0.50 (−0.96, 1.05)	0.767
**Steps**	4643 ± 3846	4477 ± 3225	6755 ± 3821	6597 ± 3333	2230 (1012, 3348) **	2085 (830, 3339) **	0.868
**VPAD, minutes**	2.41 ± 2.1	2.45 ± 1.2	2.36 ± 1.4	2.10 ± 2.0	−0.45 (−0.62, 0.52)	−0.35 (−0.95, 0.25)	0.460
**VPAH, minutes**	2.00 ± 1.2	2.20 ± 0.9	2.23 ± 0.9	2.45 ± 1.2	0.23 (−0.12, 0.58)	0.25 (−0.12, 0.62)	0.928
**TV viewing, hours**	0.33 ± 0.5	0.45 ± 0.5	0.5 ± 0.5	0.55 ± 0.5	0.18 (−0.70, 0.42)	0.17 (−0.16, 0.36)	0.672
**Video gaming, hours**	0.95 ± 0.2	0.85 ± 0.4	0.90 ± 0.3	0.90 ± 0.3	−0.05 (−0.21, 0.12)	0.05 (−0.12, 0.22)	0.402
**Internet use, hours**	0.41 ± 0.5	0.55 ± 0.5	0.52 ± 0.5	.40 ± 0.5	0.10 (−0.09, 0.30)	−0.15 (−0.35, 0.05)	0.079 ^a^
**Competence**	5.53 ± 0.9	5.55 ± 1.1	5.97 ± 0.7	5.6 ± 1.4	0.43 (0.12, 0.74) **	0.50 (−0.26, 0.38)	0.093
**Amotivation**	2.27 ± 1.0	2.26 ± 0.9	2.34 ± 0.8	2.06 ± 1.2	0.68 (−0.27, 0.41)	−0.20 (−0.55, 0.15)	0.275
**External**	2.00 ± 0.5	1.77 ± 0.4	2.14 ± 0.4	1.83 ± 0.6	0.14 (−0.78, 0.35)	0.62 (−0.16, 0.29)	0.632
**Introjected**	1.83 ± 0.8	1.78 ± 0.6	2.14 ± 0.6	1.87 ± 0.8	0.31 (0.04, 0.58) *	0.87 (−0.20, 0.37)	0.264
**Identified**	1.38 ± 1.1	0.98 ± 0.9	1.48 ± 1.0	1.11 ± 1.0	0.10 (−0.22, 0.43)	0.13 (−0.21, 0.46)	0.923
**Integrated**	2.22 ± 0.6	2.15 ± 0.6	2.51 ± 0.6	2.11 ± 0.8	0.26 (0.02, 0.57)	−0.04 (−0.33, 0.25)	0.102
**Intrinsic**	2.23 ± 0.5	2.22 ± 0.5	2.39 ± 0.4	2.06 ± 0.9	0.16 (−0.38, 0.06)	−0.16 (−0.05, 0.37)	0.037 ^†^
**Body appreciation**	3.25 ± 0.9	3.24 ± 1.0	3.69 ± 0.7	3.13 ± 1.1	0.44 (0.19, 0.66)	−0.11 (−0.36, 0.14)	0.003 ^†^
**Self-esteem**	3.5 ± 0.5	3.33 ± 0.4	3.51 ± 0.4	3.33 ± 0.4	0.01 (−0.13, 0.15)	0.00 (−0.15, 0.15)	0.912

^†^ Interaction term significant at *p* < 0.05. Within-group difference between baseline and post-intervention significant at * *p* < 0.05; ** *p* < 0.01; *** *p* < 0.0001. ^a^ Outcome from analysis using log10 transformed data. Statistics from linear mixed models. *Abbreviations:* VPAH: hours spent in vigorous physical activity per week; VPAD: days spent in vigorous physical activity per week.

**Table 5 children-08-00076-t005:** Participant quotes from qualitative interviews.

**Impact**		
Physical benefits of the programme?	*The first set of [fitness tests] I couldn’t run, I couldn’t do anything, and so the bleep test felt so hard and then at the end I got further and it was easier. When you’re doing it you don’t feel like you are getting better but when you look at the scores from the start and the end it makes you think that you’ve actually gotten better at it. [P19]* *I have become a lot more used to exercising different types and my stamina has improved a lot and that has definitely helped my legs are more toned [P16]*	
Psychological benefits of the programme?	*Doing [PA] more and more and more I gradually enjoyed it more and more…I definitely got more confident in myself I have definitely noticed a change in how I see myself and look at myself [P24]* *It made me less lazy and want to move more I would choose to walk the dog without being asked now [P29]*	
**Adherence**		
What facilitated adhering to the programme?	*Live workouts*	*I found (the live workouts) really good, they were all different every time so it’s not like they are repeating it’s kind of like a surprise and it was really good with the timer your ears are ready, your muscles are getting tired and you are just waiting for that beep so you can collapse [P42]* *When we’re watching [the live workout] you know it’s real, we’re looking the same, but on YouTube it’s people who are fake, it looks so easy for them but they are probably doing it like stop recording, put their make-up back on and then do it again but when we do it it’s like 100% real [P1]*
	Social support	*If it wasn’t for (the programme) I wouldn’t have done anything in lockdown at all but because of this my mum was forcing me to go out for a run and now I’m doing runs quite frequently [P19]* *To know that there is people our own age doing the same thing and feel part of a community even if we don’t know each other it’s kind of a reassurance not everyone saying to themselves “oh no everyone is going to be looking at me” cause they’re not, we’re just doing our exercise [P42]*
	Organisation and routine	*The wee table that you could put all your exercises on that definitely helped because I could be more organised and doing (exercise) more I gradually enjoyed it more [P24]* *I think I felt more motivated cause I’m keep scheduling and keep the same every week it was like I had something I needed to get done [P40]*
	Autonomy	*I liked the independence it gave you … All the exercises were really fun cause you could choose what exercises you want to do which again is really good [P42]* *In the beginning I picked all (the PA options) to try them all out and then after a couple of weeks I picked the ones I enjoyed most [P24]*
	Activity Mentors	*I think if I didn’t have that phone call or anyone to catch up with I would have just been like “meh” because of the call I stuck to everything I said I would do for that week [P25]* *Having (the activity mentor) there saying just do what you can it was good, it wasn’t like you have to do these three options or else you were kicked off (the programme) it was more like just do what you can type thing [P19]*
What were the barriers to adherence?	*The only times I didn’t do (the PA sessions) was if I had period pains [P16]* *Maybe some more (PA options) because it sometimes felt like I was doing the same ones again if I didn’t like one of them [P13]* *I think make a point you don’t need your camera on because I think if I hadn’t of known that I didn’t need my camera on then I wouldn’t have done it [P13]* *Maybe like different times (for the live workouts) probably earlier in the morning, it’s a difficult time like 6pm is such a busy time at the end of the day [P1]*	

## Data Availability

Data presented in this study are available on request from the corresponding author. The data are not publicly available due to privacy.

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
