# Peer review of "Formative Evaluation of a Home-Based Physical Activity Intervention for Adolescent Girls—The HERizon Project: A Randomised Controlled Trial"

_children, 2021, doi:10.3390/children8020076_

Round 1

Reviewer 1 Report

children-1034846

Formative Evaluation of a Home-based Physical Activity Intervention for Adolescent girls - The HERizon Project: A Randomised Controlled Trial

The paper presents important results from a formative evaluation of a Physical Activity intervention for adolescent girls. The study has a strong design and provides an interesting view about a project literally cached up during the Covid 19 lockdown. I’m glad to the opportunity to read the paper and also make some small contributions.

The manuscript is well written and I have some (few) specific suggestions to improvement.

  1. Abstract: It would be good to briefly describe, how  the secondary outcomes were assessed since authors only stated that PA was self-reported.
  2. Methods: It is not clear why the number of 40 participants was aimed
  3. Please correct line #310: analysis of variance (ANVOCA) please change to (ANCOVA)
  4. Also, in the methods description, it was not clear how the content of the interviews was analyzed, if they were recorded and if any software was used to support qualitative analysis and the if the categories emerged from data or were considered a priori

Moreover, I would like to suggest Authors to consider a brief comment on the need to use an easy but more generalizable way to measure cardiorespiratory fitness in home-based programs.

Reviewer 2 Report

As the groups are randomized any differences in baseline measures are random so therefore it probably isn't necessary to include p-values in Table 3.

The high values of quite a few of the SD's in Table 4 imply that some of the data is skewed so may not be linear.  Were distributions checked? 

WRT the models, please note the BMJ paper by Vickers and Altman: Analysing controlled trials with baseline and follow up measurements (BMJ 2001323 doi: https://doi.org/10.1136/bmj.323.7321.1123).

A better option would be to run it as a longitudinal model (LMM or GLMM) - which will also increase power - to obtain the estimated mean differences between timepoints.   If there is some transformation required then it would be better to run GLMM's so that the resulting marginal means are correct. 

I am concerned with the number of covariates adjusted for in the model- was there enough power?  Better to maximise the existing power of this small sample by using longitudinal models.  There may be more significant differences with more power and will also enable a more accurate effect size estimation for a primary outcome

Reviewer 3 Report

A few suggestions comments that need to be addressed.

Intro: I would recommend line 77 is moved to line 88 to improve readability.

Methods

I am unsure of the detail and would welcome additional info as to how reliable the fitness measures were. Did you consider recording the participants via zoom etc. to ensure compliance? If so, why was this not done, particularly as this was the primary outcome. This would be methodologically sound, particularly for the push up test.

Whilst adapting due to COVID and completing a study is to be commended, we have to be careful when adapting potentially impacts the reliability of the asessmenrs.

How did you ensure participants measured 20 m or the standing long jump accurately? Was equipment loaned/ encouraged etc.

Table 2 needs formatting (adolescents)

I feel the physical activity test results require greater discussion. No increase in MVPA but significant improvements in tests. How can this be attributed to the intervention alone. Were the participants doing something else or did the control group do nothing (Were schools closed, here students in lockdown?).

The robust design has led to good feasibility which is pleasing when looking for research that addresses such a key area. As is the rate of compliance. This is clearly a study strength.

Whilst the limitation of the PA tests is discussed I think this needs to be discussed in greater detail.

Reviewer 4 Report

This paper examined the feasibility and preliminary effectiveness of an RCT that aims to improve PA in adolescent girls. The paper is very well written. There are only minor typographical errors throughout (e.g. no space between values and units). I have a few queries below – mostly about the methodology and analysis.

L128: What do you mean by country-level stratification? What countries?

L308: Please state explicitly that the assumptions for these statistical methods were checked (and met). Same for the regression models.

L310: The ANCOVA acronym is incorrect.

L310–314: Is there a reason why baseline scores were added as fixed effects when using the change score as the dependant variable? Several studies have shown this approach may inflate regression coefficients. Did you perform a sensitivity analysis where post-intervention is used at the dependant variable?

L314–315: I don’t think you should state this analysis was exploratory. You had explicit aims. Perhaps you can simply say this was a feasibility study with a small sample size, as a rationale for using LSD.

L366–369: This analysis was not described in the methods section.

Table 4: The footnote suggests there are paired t-test results presented somewhere in this table, but I cannot see any.

I acknowledge this is a feasibility study, but I encourage the authors to provide a statement about the sample size. The authors have stated that 42 participants is acceptable for examining the study aims – please provide a rationale for this. You may also consider presenting an effect size statistic (i.e. eta-squared for your ANCOVA models) to assist other researchers when estimating sample sizes for future studies.
